# Expansion of myeloid suppressor cells and suppression of Lassa virus-specific T cells during fatal Lassa fever

Blaise Lafoux[1,2], Gustave Fourcaud[1,2], Jimmy Hortion[1,2], Laura Soyer[1,2], Alexandra Journeaux[1,2], Clara Germain[1,2], Stéphanie Reynard[1,2], Hadrien Cousseau[1,2], Clémentine Larignon[1,2], Natalia Pietrosemoli[3], Séverine Croze[4], Joël Lachuer[4], Emeline Perthame[3], Sylvain Baize [1,2]*

1 Unité de Biologie des Infections Virales Emergentes, Institut Pasteur, Université Paris Cité, Lyon, France, 2 Centre International de Recherche en Infectiologie, Université Claude Bernard Lyon 1 (UCBL), Institut National de la Santé de la Recherche Médicale (INSERM), Ecole Normale Supérieure de Lyon, Centre National de la Recherche Scientifique (CNRS), Lyon, France, 3 Bioinformatics and Biostatistics Hub, Institut Pasteur, Université Paris Cité, Paris, France, 4 ProfileXpert, SFR Santé Lyon-Est, UCBL, CNRS, INSERM, Lyon, France

* sylvain.baize@pasteur.fr

## Abstract

Lassa fever is a highly lethal hemorrhagic fever endemic to West Africa. In the absence of efficient prophylactic or therapeutic countermeasures, it poses a substantial threat to public health in this region. The pathophysiological mechanisms underlying the severity of the disease are poorly known because Lassa virus (LASV), its causative agent, has to be handled in BSL-4 laboratories and access to clinical samples is difficult. The control of Lassa fever is associated with a rapid and well-balanced immune response and viral clearance. However, severe disease is characterized by uncontrolled innate immune activation and symptoms reminiscent of sepsis and a cytokine storm. In a model of cynomolgus monkeys infected with two different strains of the virus, one causing moderate disease and the other a lethal outcome, we show that the control of LASV infection is characterized by the induction of a LASV-specific T-cell response, whereas severity is associated with the expansion of suppressive myeloid cells, alterations of the stromal network of secondary lymphoid organs, and the anergy of specific T cells. These results suggest that T cells are crucial for the control of LASV and that immunomodulatory therapeutics, such as checkpoint inhibitors, could contribute to new therapeutic strategies to treat Lassa fever. They also highlight how immunosuppressive mechanisms described in sepsis and cancer patients may play a role in the pathogenicity of Lassa fever, as well as in other similar hemorrhagic fevers.

**Data availability statement:** We have deposited our transcriptomic data at the following URL: https://zenodo.org/records/14673870. All other relevant data are in the manuscript and its supporting information files.

**Funding:** This study was funded by a French government grant (France 2030 program) operated by ANRS-MIE (ANRS-23-PEPR-MIE 23465) to SB. BL was a recipient of a PhD grant funded by Lyon 1 University (ED 340 BMIC). GF and JH received a PhD grant from the Ecole Normale Supérieure de Lyon and HC a PhD grant from the Ecole Normale Supérieure de Paris. LS received a PhD grant from ANRS-MIE (ANRS-23-PEPR-MIE 23465). The funders had no role in study design, data collection and analysis, decision to publish, or preparation of the manuscript.

**Competing interests:** The authors have declared that no competing interests exist.

## Author summary

The immune and pathological mechanisms by which some Lassa fever patients are able to limit Lassa virus spreading and to recover whereas others experienced systemic dissemination and catastrophic illness are still unclear. Here, we demonstrated using a relevant animal model that non-fatal Lassa fever is accompanied by the induction of LASV-specific T-cell responses that probably largely account for the control of viral replication. In contrast, we showed that no specific T cells are generated during fatal infection, and that this is due to immunosuppression mediated by myeloid suppressive cells and by severe alterations of secondary lymphoid organs. The significance of this work may extend to other viral hemorrhagic fevers and pave the way to new therapeutic targets based on immunomodulation.

## Introduction

Lassa fever is a zoonotic viral hemorrhagic fever (VHF) caused by the Lassa virus (LASV), a member of the *Arenaviridae* family. This disease is endemic to West Africa, where it causes thousands of cases per year, mainly in Nigeria, Sierra Leone, Guinea, and Liberia [1–3]. It is estimated that 9% of the population in the endemic region has already been in contact with the virus and expansion of the area of endemicity is expected over the next several decades, putting an increasing number of people at risk [4–6]. Consequently, Lassa fever has been placed by the World Health Organization on the list of priority pathogens requiring urgent development of countermeasures. The main reservoir host of the virus is the rodent *Mastomys natalensis*, present in all of sub-Saharan Africa [7]. It has been postulated that the virus is transmitted to humans by the inhalation of viral particles contained in the urine and feces of infected rodents, by the consumption of contaminated food, or while providing medical care to infected patients. While most infected individuals appear to remain asymptomatic and are not diagnosed, the case-fatality ratio for confirmed cases ranges between 15 and 30% [4,8]. The first phase of the disease is characterized by non-specific clinical signs, such as fever, fatigue, sore throat, and muscle pain, and can be easily confused with those of other endemic pathogens. After a few days, additional severe symptoms can arise, such as face and neck edema, bleeding, dyspnea, seizures, and the loss of consciousness. Death generally occurs 1–6 days after hospitalization and is preceded by multiorgan failure, likely caused by distributive shock [8–10]. The treatment of infected patients relies on supportive therapy, including fluid resuscitation, prophylactic courses of antibiotics, blood transfusions, and renal replacement therapy. Several recombinant vaccines expressing viral antigens have undergone phase I clinical trials, but none have yet been approved for use in patients [11–13]. The burden of this disease is increased by the frequent neurological sequelae observed among survivors, as 13–29% report sensory-neural hearing loss [14]. The severity of this disease, the absence of efficient countermeasures, and the threat that it poses to human

PLOS Pathogens

populations have led to the classification of LASV as a risk group 4 pathogen. This constraint, in addition to the difficulty of obtaining samples from infected patients, has hindered research on Lassa fever. Indeed, very little is currently known about the pathogenesis of this infection and the mechanisms associated with survival and death. Studies on infected patients have reported a link between severe outcomes and several markers or symptoms associated with organ failure (abnormal vital signs, renal markers, hepatic enzymes), hemostasis defects (bleeding, edema, plasminogen activator inhibitor 1, thrombomodulin), and inflammation (TNF receptor 1, IL-8, CCL2) [8,15]. Survival, on the other hand, appears to depend on the efficiency of cell-mediated adaptive immunity. Indeed, convalescent patients show an antigen-specific, polyfunctional, and long-lived T-cell response [16–21]. In the relevant cynomolgus monkey model of infection, the control of viral replication correlates with the expansion of activated proliferating T cells in the blood, whereas LASV-specific IgGs are secreted both in lethal and non-lethal outcomes and neutralizing titers rise only days after viral clearance [22]. Secondary lymphoid organs are an early major reservoir for LASV infection whatever was the severity of the disease [23]. On the contrary, a systemic viral dissemination that leads to massive infiltration of macrophages and neutrophils and to excessive inflammation is observed only during fatal infection. The immune response of severely ill individuals or of fatally-infected non-human primates (NHP) appears to be paradoxical. Innate immunity is overly activated, as shown by the infiltration of monocytes and neutrophils in the tissues, upregulation of activation markers on circulating immune cells, and high plasma levels of pro- and anti-inflammatory cytokines, such as IL-6, TNF-α, IL-10, and IL-1RA [22–26]. On the other hand, the cell-mediated adaptive response appears to be defective in lethal outcomes, with low expression of activation and proliferation markers on circulating T-cells and high numbers of apoptotic T-cells [22]. Bystander activation of T-cells has also been proposed as a mechanism for LASV-induced pathogenesis and tissue damage [26]. We used two different strains of LASV, Josiah and AV, to induce lethal and non-lethal outcomes, respectively, in cynomolgus monkeys that mimic human Lassa fever [22,23]. By performing sequential euthanasia during the early (2 days post-infection [DPI]), intermediate (5 DPI), late (11 DPI), and convalescent stages (28 DPI) of the infection, we have gained insights into the complex immune populations that infiltrate the organs. In an effort to reduce the number of primates subjected to experimentation, we used biological material generated during this study to provide evidence that recovery from acute Lassa fever is associated with an LASV-specific CD4+ and CD8+ T-cell response, whereas fatal Lassa fever is characterized by the expansion of immunosuppressive myeloid cells, alterations of the stromal network of lymph nodes (LNs), and suppression of the virus-specific T-cell response.

## Results

### The control of LASV infection is associated with a virus-specific T-cell response

We used samples from cynomolgus monkeys infected with a lethal (Josiah) or non-lethal (AV) strain of LASV to compare the functionality of the cell-mediated immune response according to the outcome of the disease. The clinical signs observed in these animals after AV and Josiah Lassa virus infection, as well as the survival curves, have been previously described and are not presented here [22]. We measured the frequency of virus-specific T cells in the blood and secondary lymphoid organs (SLOs) at various time points post-infection. We stimulated T cells using a pool of overlapping peptides spanning the entire sequence of the LASV glycoprotein complex (GPC) and nucleoprotein (NP) before staining for several surface markers and cytokines. The peripheral blood mononuclear cells (PBMCs) of AV-infected animals showed increase frequency of LASV-specific CD4+ T cells expressing IFNγ and/or CD154 and CD8+ T cells expressing IFNγ and/or CD137, starting at 12 DPI and culminating at 28 DPI (Fig 1A). A transient decrease of the percentage of specific CD8+T cells was nevertheless observed at 18 DPI in these animals. By contrast, we did not observe any increase in the percentage of virus-specific T-cells positive for IFN-γ in Josiah-infected macaques, but rather a moderate increase in the percentage of stimulated CD4+ CD154+ T cells at 12 DPI and a stronger but transient increase in that of the CD8+ CD137+ population at 3 DPI. In the SLOs of AV-infected animals, we observed an increase in the percentage of virus-specific CD4+

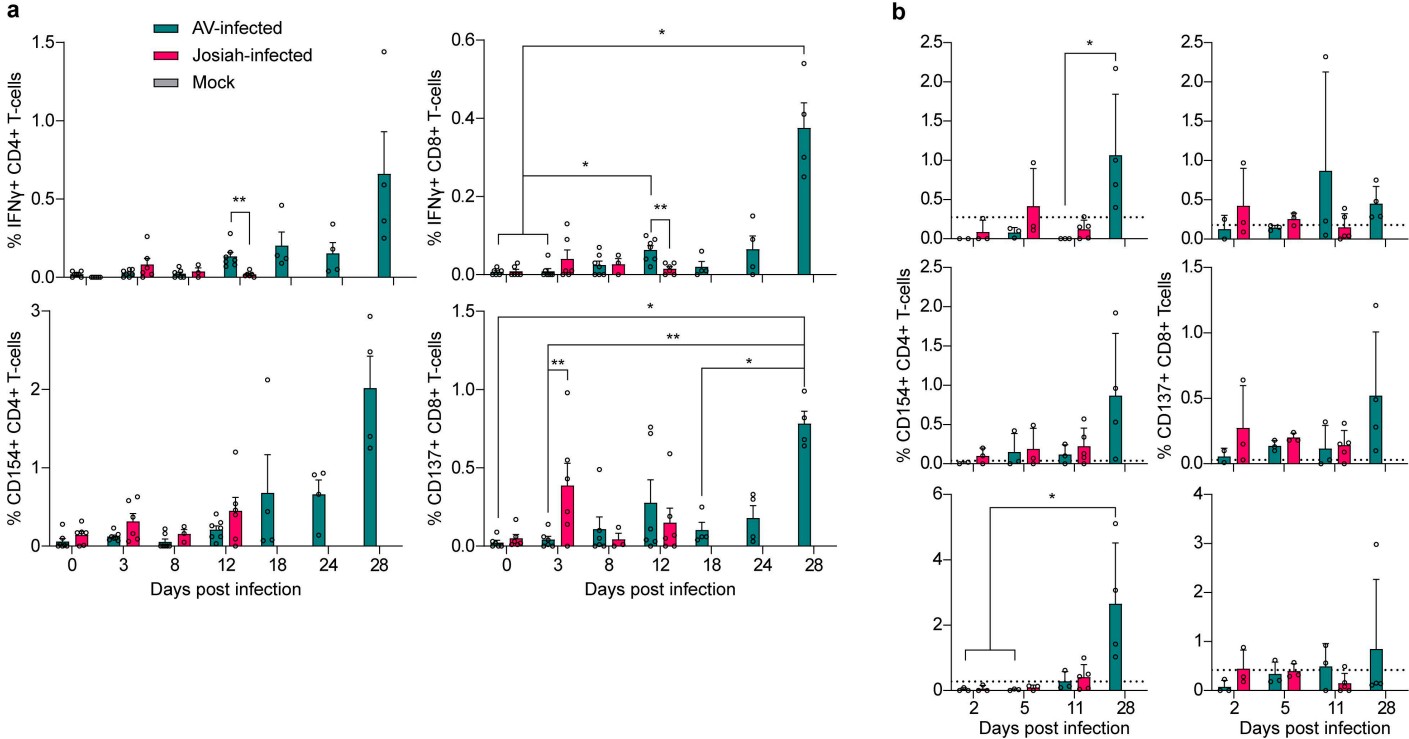

**Fig 1. Virus-specific T-cell response in the PMBCs and SLOs of infected animals. A.** Percentage of CD4+ (left column) and CD8+ (right column) T cells from PBMCs expressing the activation markers IFN-γ (top row) or CD154/ CD137 (bottom row) in response to antigen stimulation with viral peptides. Data are presented as individual values, with the mean and standard error of the mean (SEM) of 6 Josiah-infected animals and 7 AV-infected animals until 12 DPI and 4 AV-infected animals after 12 DPI. Differences between the AV and Josiah groups for the same time point were tested using an unpaired non-parametric Mann-Whitney test and differences between time points within the same group were tested using mixed-effects ANOVA with repeated measures (Tukey's test) and the Geisser-Greenhouse correction. For all figures, asterisks signal a p-value < 0.05 (*), < 0.01 (**), < 0.001 (***) or < 0.0001 (****). **B.** Percentage of CD4+ T cells expressing CD154 (left column) and CD8+ T cells expressing CD137 (right column) for cells obtained from the inguinal LN (top row), mesenteric LN (middle row), or spleen (bottom row) in response to antigenic stimulation. The dotted black line corresponds to the mean of the non-infected animals (n = 3). Data are presented as individual values and the mean and SEM of n = 3 "Josiah 2 DPI", n = 3 "Josiah 5 DPI", n = 5 "Josiah 11 DPI", n = 3 "AV 2 DPI", n = 3 "AV 5 DPI", n = 3 "AV 11 DPI" and n = 4 "AV 28 DPI". There is one missing sample for both LNs in the "AV 2 DPI" group. Differences between time points within the same group were tested using an unpaired non-parametric Kruskal-Wallis test with Dunn's multiple comparisons test.

and CD8+ T cells expressing CD154 and to a lesser extent CD137, respectively, most notably at 28 DPI (Fig 1B). On the other hand, we did not observe any significant increase in T-cell activation following antigenic re-stimulation in the spleen or inguinal or mesenteric LNs of Josiah-infected individuals.

### The lethal strain of LASV targets immunoregulatory macrophages during early infection

We compared the tropism of LASV AV and Josiah in the draining inguinal LN (ILN). Indeed, ILNs are in our model the site where viral amplification takes place after the first cycles of replication at the inoculation site [22]. We performed multiplexed immunofluorescence (IF) and in-situ hybridization (ISH) targeting the LASV genome on histological slides of the ILN that drains the site of inoculation. We generated whole-slide images by tiling individual fields of view obtained at high magnification before unmixing the spectra from overlapping fluorophores (Fig 2A). We then used QuPath software to automatically detect the cell boundaries and nuclei and classify each cell according to its positivity for the markers. Between 300 and 17,000 cells were classified as infected. The viral genome was associated with macrophages (CD68+ DC-SIGN-), dendritic cells (DCs) (DC-SIGN+ CD68-), immunoregulatory macrophages (CD68+ DC-SIGN+) [27–29], endothelial cells (CD31+), and

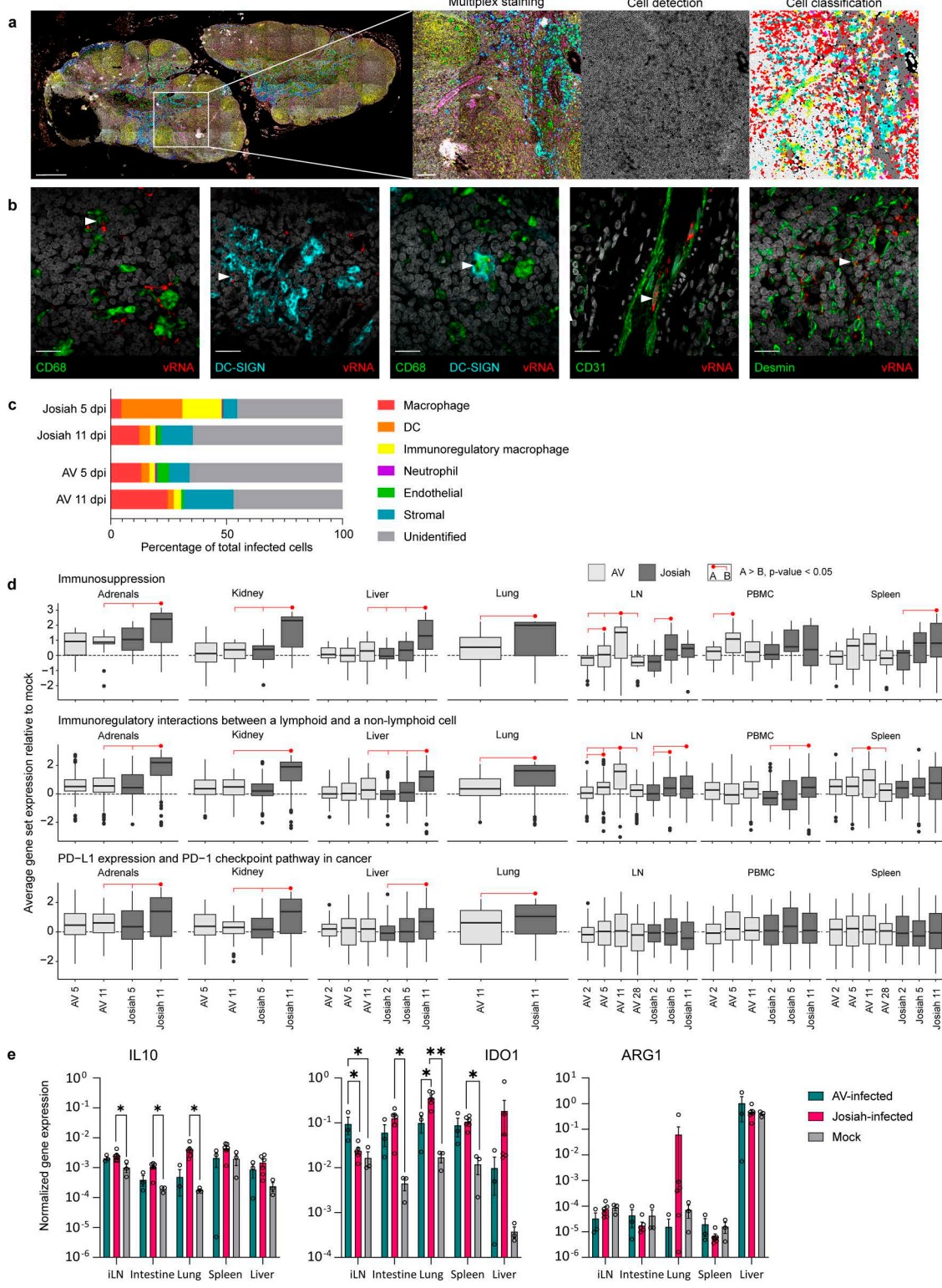

**Fig 2. Tropism of LASV Josiah and AV in the LN and modifications of gene expression associated with immune regulation in the organs of infected animals. A.** Illustration of the workflow of the imaging and analysis steps used for multiplexed histology. First inset: mosaic reconstitution of a whole slide image obtained from hundreds of individual fields by confocal microscopy. Scale = 500 μm. Second inset: higher magnification of the white square

showing the merge of seven fluorescent channels. Scale = 100 μm. Third inset: automated cell detection based on 4′,6-diamidino-2-phenylindole (DAPI) staining. Fourth inset: automated classification of detected cells according to their estimated positivity for each fluorescent marker. **B.** High-magnification insets illustrating the different types of infected cells in the LNs. White arrowheads point towards cells expressing a phenotypic marker and containing viral RNA. **C.** Proportion of each identified cell type among the total infected cells. Cell types are defined as follows: macrophage = CD68⁺ DC-SIGN⁻ CD31⁻ desmin⁻, DC = DC-SIGN⁺ CD68⁻ CD31⁻ desmin⁻, immunoregulatory macrophage = CD68⁺ DC-SIGN⁺ CD31⁻ desmin⁻, neutrophil = calprotectin⁺ CD68⁻ DC-SIGN⁻ CD31⁻ desmin⁻, endothelial = CD31⁺ CD68⁻ DC-SIGN⁻ calprotectin⁻, and stromal = desmin⁺ CD68⁻ DC-SIGN⁻ calprotectin⁻. Data correspond to the mean of three animals per time point and per viral strain. **D.** Standardized average gene expression of three relevant gene sets in the adrenals, kidneys, liver, lungs, LNs, PBMCs, and spleen harvested at different times post-infection. Data are normalized against the average expression in the non-infected condition. They are represented by their interquartile range (IQR, top and bottom limits of the boxes), their maximum and minimum (Q1 - 1.5 * IQR or Q3 - 1.5 * IQR, range of the vertical black lines), and their median (horizontal black line). Strains and time points were compared using a linear mixed model, with a fixed model adjusted for the group and a random structure adjusted for the gene ID to account for the dataset structure. The significance is represented by the red dots and lines. **E.** Measurement of the expression of three relevant genes in different organs at 11 DPI by RT-qPCR. Values were normalized against the expression of the housekeeping gene GAPDH. Data are presented as individual values and the mean and SEM. Differences between groups for the same gene and same organ were tested using a one-way ANOVA or a Kruskal-Wallis, test depending on the normality of the distribution according to a D'Agostino-Pearson test. For **D** and **E**, n = 3 animals per group and time point, except for "Josiah 11 DPI" (n = 6) and "AV28 DPI" (n = 4).

stromal cells (desmin⁺) in the LN (Fig 2B). At 5 DPI, Josiah LASV was most frequently associated with DCs and immuno-regulatory macrophages, with a small percentage of classical macrophages and stromal cells (Fig 2C). On the other hand, the AV strain was present mostly in macrophages and endothelial and stromal cells. At 11 DPI, the cell tropism of the two strains became similar, mostly represented by macrophages and stromal cells. We determined whether immunoregulatory pathways were upregulated during lethal infection by performing transcriptomic analysis of the organs of infected animals. We observed the upregulation of several gene sets associated with immune suppression and regulation specifically in the adrenals, kidneys, liver, lungs, and PBMCs of Josiah-infected animals, culminating at 11 DPI (Figs 2D, S1 and S2). Surprisingly, the upregulation of these genes was most notable at 11 DPI in the LNs and spleen in the AV group, even though it was also observed at 5 DPI in the Josiah group. We confirmed by RT-qPCR that the immunosuppressive genes *il10* and *ido1* were upregulated in the lungs and intestines of Josiah-infected animals versus the mock- and AV-infected animals (Fig 2E). In ILN, *il10* expression was higher in Josiah-infected animals than in mock-infected animals, whereas *ido1* expression was higher in AV-infected animals. In spleen and liver, only *ido1* was differentially expressed, with upregulation of its synthesis in Josiah-infected animals, and to a lesser extent in AV-infected ones, compared to mock-infected animals.

## Primary human macrophages show a stronger response to infection with AV

To determine whether the immunoregulatory phenotype observed for Josiah-infected macrophages could be viral strain-dependent, we infected primary human macrophages derived from PBMCs with LASV Josiah or AV. Total RNA was extracted at 24 hours post-infection (hpi) and surface marker expression was studied at 48 hpi. Both viral strains induced an increase in the percentage of CD274 + and of HLA-DR⁻ macrophages, whereas the expression of the M2 polarization marker CD163 was not modified (S3A Fig). Transcriptome analysis showed gene expression to be considerably more altered in terms of the number of genes and the intensity of expression in macrophages infected with AV than with Josiah (S3B Fig). Most pathways classically associated with the macrophage response to viral infection were upregulated in both conditions (MHC-I, M1-differentiation, IFNα, interferon-stimulated genes, and IFNγ) but genes associated with MHC-II antigen presentation were downregulated in infected macrophages (S3C Fig). There was overall upregulation of genes associated with cytokines, inflammation, and immune signaling for both viral strains, as well as significant overexpression of suppressive genes, such as *ido1*, *cd274*, *tgfb1*, *il10ra*, and *il10rb*, and increased expression of suppressive pathways, such as IL-10 and TGFβ signaling, immunoregulatory interactions between lymphoid and non-lymphoid cells, programmed cell death, and apoptosis (S3D-F Fig).

## Immunosuppressive myeloid cells expand during lethal infection

We investigated the potential role of myeloid-derived suppressor cells (MDSCs) in the disease. MDSCs are immature myeloid cells that expand in pathological immune responses, such as cancer or persistent viral infections, and they

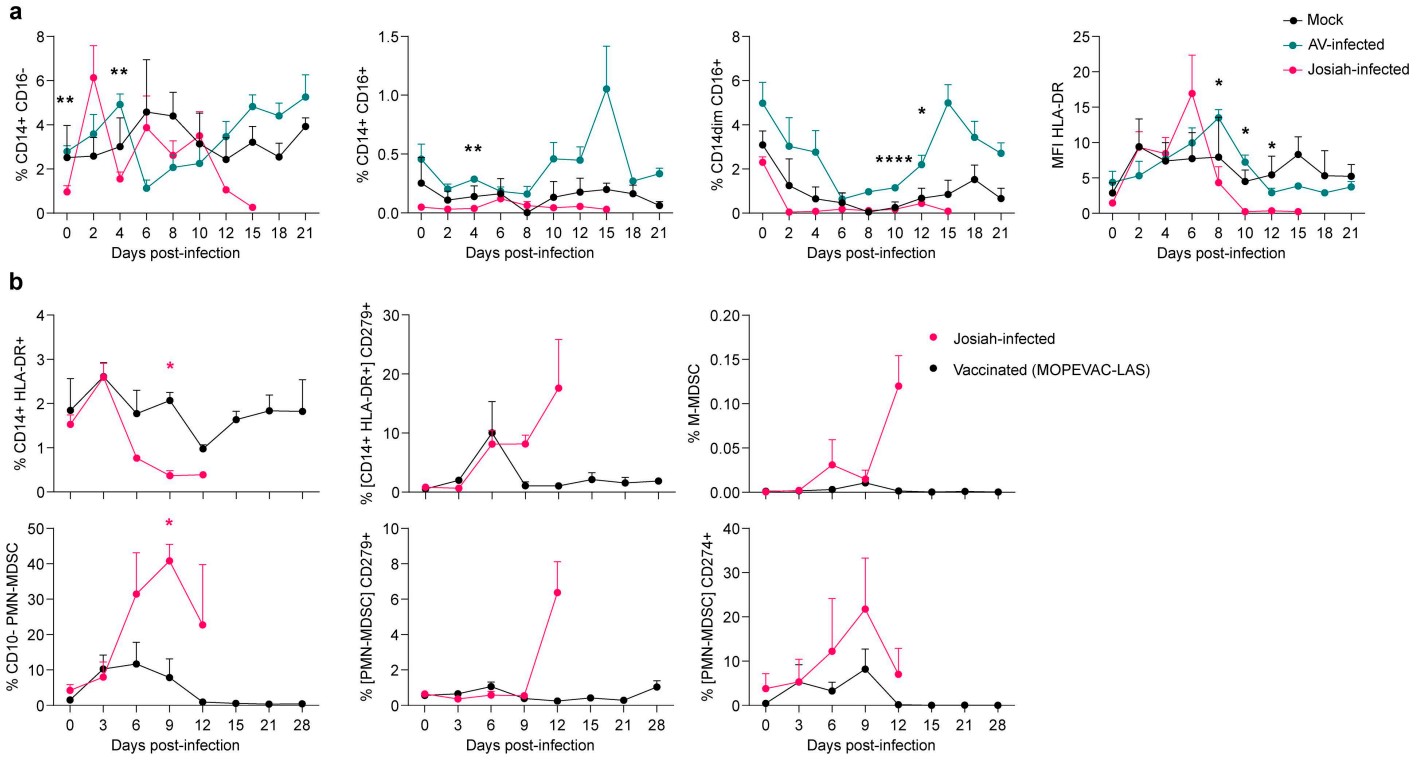

**Fig 3. Longitudinal characterization of monocytes and MDSCs in the blood. A.** Frequency of different types of mature monocytes (classical [CD14+ CD16-], intermediate [CD14+ CD16+], and non-classical [CD14- CD16+]) relative to the total leukocyte number in the blood and median fluorescence intensity of HLA-DR for the sum of the three monocyte subpopulations, in arbitrary units. Data are presented as the mean and standard error of the mean (SEM) of 3 mock-infected animals, 4 AV-infected animals, and 5 Josiah-infected animals. **B.** MDSC phenotypes and functional markers in the blood. Data correspond to the percentage of the total leukocyte number, except when a population of interest is specified between brackets. MDSC subtypes are defined as follows: M-MDSC=PBMC morphological gate, CD14+, HLA-DR-, CD11b+, CD33+ and PMN-MDSC=neutrophil morphological gate, CD14-, HLA-DR-, CD66abce+, CD11b+. Data are presented as the mean and SEM for 3 vaccinated controls and 3 unvaccinated Josiah-infected animals. For **A** and **B**, asterisks correspond to the p-value of a mixed-effects ANOVA with repeated measures and Geisser-Greenhouse correction comparing 3 groups (Tukey's test) or 2 groups (Sidak's test) for the same time point. Asterisks indicate a significant difference: blue for "AV - Mock", pink for 'Josiah - Mock', and black for "AV - Josiah".

suppress T-cell responses. The frequency of mature classical monocytes (CD14+ CD16-) in the blood of Josiah-infected animals decreased at late stages (12–15 DPI), whereas it increased in the AV group (Fig 3A). The frequency of intermediate (CD14+ CD16+) and non-classical monocytes (CD14dim CD16+) was also higher in non-lethal disease than for the Josiah- and mock-infected animals. The mean fluorescence intensity of the major histocompatibility complex (MHC) class II molecule HLA-DR decreased for the monocytes of Josiah-infected animals as soon as 8 DPI and became undetectable by 10 DPI, whereas it increased before returning to baseline at 12 DPI in the AV group. In another cohort of animals, we confirmed that the frequency of CD14+ HLA-DR+ monocytes decreased following Josiah infection from 6 DPI to euthanasia relative to vaccinated healthy controls (Fig 3B). The percentage of these monocytes expressing CD279 was also higher for Josiah-infected animals at 9 and 12 DPI. We observed a late moderate increase in the frequency of monocytic MDSCs (M-MDSCs) (CD14+, HLA-DR-, CD11b+, CD33+) in this group, and, more strikingly, the strong expansion of CD10- polymorphonuclear MDSCs (PMN-MDSCs) (CD14-, HLA-DR-, CD66abce+, CD11b+) from 6 DPI, reaching a maximum of 40% of total leukocytes at 9 DPI. These cells were also more frequently positive for CD279 and CD274 (PD-L1) in the Josiah group than those from vaccinated controls.

We next performed multiplexed IF on tissues harvested at 2, 5, or 11 DPI to determine whether these suppressive myeloid cells were also present in the organs (Fig 4A). We manually annotated the follicular and paracortical zones of the

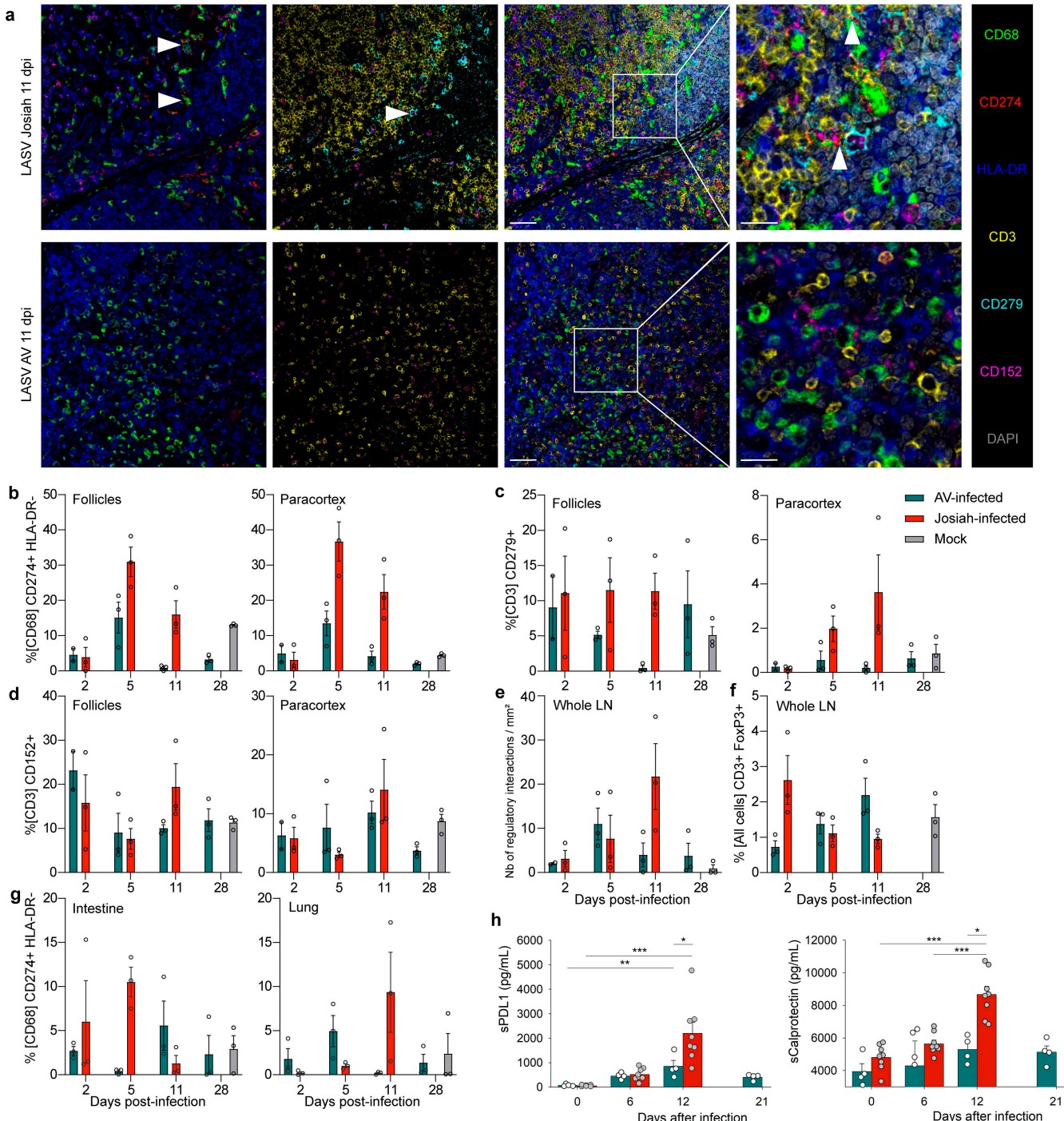

**Fig 4. Histocytological characterization of the suppressive environment of the LNs and quantification of soluble PD-L1 and calprotectin in plasma. A.** Confocal images captured from the paracortex of the LNs of infected animals successively showing CD274+ macrophages (first column, white arrowheads), CD279+ T cells (second column, white arrowhead), and cell-cell contacts between these two cell types (third and fourth columns, white arrowheads). Column one shows the merge of the markers CD68, CD274, and HLA-DR; column two shows the merge of the markers CD3, CD279, and CD152; column three shows the merge of all six markers with DAPI, and column four shows the higher magnification of the white squares. Scale of columns 1-3 = 50 μm and column 4 = 20 μm. **B-D** Percentage of CD274+ HLA-DR- cells among total macrophages **(B)**, CD279+ cells among total T-cells **(C)**, and CD152+ among total T cells **(D)** in the two main histological zones of the LN. **E.** Number of CD279+ T cells for which the distance to a CD274+ macrophage is less than the sum of the mean radius of a macrophage, the mean radius of a T cell, and the standard deviation of both

these means, divided by the total area of the slide in mm$^2$. **F.** Percentage of CD3$^+$ FoxP3$^+$ cells (Tregs) among total cells in whole LNs. **G.** Percentage of CD274$^+$ HLA-DR$^-$ cells among total macrophages in the epithelium of the large intestine and in the lung. For **B-G**, data are presented as individual values and as the mean and SEM of three animals per group and time point. No statistical tests were performed because we were unable to verify the normality of the data distribution and because the euthanasia of the animals at sequential time points does not fit a repeated measures design. **H**. Soluble PD-L1 (left graph) and calprotectin (right graph) were quantified in plasma obtained at 0, 6, and 12 days after infection in AV- (blue bars and white circles, n = 4) and Josiah-infected animals (red bars and gray circles, n = 8), as well as 21 days after infection in AV-infected animals. The results are presented in pg/mL as the mean ± SEM and as individual values. Significant differences between AV- and Josiah-infected animals for a given timepoint were calculated using a Welch's t-test (* indicates $p < 0.05$). Significant differences between timepoints within groups of animals were calculated using a one-way Anova test, with * indicating $p < 0.05$, ** indicating $p < 0.001$ and *** indicating $p < 0.0001$.

LNs before quantifying different cell populations. We observed a higher percentage of macrophages with an immunosuppressive phenotype (CD68$^+$ CD274$^+$ HLA-DR$^-$) in both histological layers of the LNs of Josiah-infected animals at 5 and 11 DPI than for those of the AV group (Fig 4B). There was also a higher percentage of T cells expressing CD279 in the paracortex of the LNs of the Josiah-infected animals (Fig 4C) but there was no striking difference in CTLA-4 expression on the T cells (CD3$^+$ CD152$^+$) between groups (Fig 4D). In addition, the calculated spatial density of direct cell-cell interactions between CD279-expressing T cells and CD274-expressing macrophages was the highest in the LNs of Josiah-infected animals at 11 DPI (Fig 4E). The frequency of regulatory T cells (Tregs) (CD3$^+$ FoxP3$^+$) was higher at 2 DPI in Josiah-infected animals than AV, and an inverse tendency was observed at day 11, but these differences were not significant and the percentage of Tregs in infected monkeys remains close to the one observed in mock-infected animals (Figs 4F and S4). Finally, a higher percentage of immunosuppressive macrophages (CD68$^+$ CD274$^+$ HLA-DR$^-$) was measured in intestine and lungs of Josiah-infected animals at 5 and 11 DPI, respectively, than for those of the AV group (Fig 4G). We observed rare cells positive for IL-10 mRNA by ISH but were unable to determine their cell type (S4 Fig). We were able to confirm that suppressive macrophages were also more frequent in the large intestine at 5 DPI and lungs at 11 DPI for the Josiah-infected animals than those of the AV and mock groups (S5 Fig). Levels of soluble (s)PD-L1 were significantly increased in the plasma from LASV-infected animals at day 12 after infection compared to the levels detected before infection, and concentrations measured in the plasma from Josiah animals were significantly higher than in AV animals (Fig 4H). Soluble calprotectin levels were significantly higher in the plasma from Josiah-infected animals at day 12 after infection compared to other timepoints and to AV-infected animals at the same time.

## The stromal network of the lymph nodes is altered in Josiah-infected animals

The loss of histological structure in SLOs is a common anatomopathological feature of severe Lassa fever and we have shown that local control of viral replication in the LNs is associated with survival. Hence, we performed multiplex immunofluorescence and ISH for the viral genome on the LNs of infected animals to study their stromal network, which is composed of several cell types expressing desmin, most notably, fibroblastic reticular cells (FRCs) throughout the organ and fibroblastic dendritic cells (FDCs) in the follicles. We manually annotated the follicles and paracortex on whole-slide images before performing the analysis. For Josiah-infected animals, we observed a higher percentage of stromal cells positive for CD274 relative to the AV and Mock groups, starting at 5 DPI and decreasing by 11 DPI, both in the follicles and paracortex (Fig 5A and 5B). There was also a higher percentage of stromal cells expressing cleaved caspase-3, which denotes undergoing apoptosis, in the paracortex of the LNs of Josiah-infected animals, starting at 5 DPI and culminating at 11 DPI, relative to their baseline and the other conditions. Interestingly, we observed a massive influx of neutrophils (calprotectin$^+$) positive for granzyme B (GrzB) in direct proximity to the infected cells (Fig 5A). We compared the percentage of positivity for these markers on individual infected or non-infected cells on the same microscope slide to determine whether CD274 expression and apoptosis were related to the infection of stromal cells. Stromal cells infected with either AV or Josiah were more frequently positive for CD274, but the infection had no notable impact on cCasp3 expression (Fig 5C). We used a model of human immortalized LN FRCs (ilnFRCs) and confirmed that

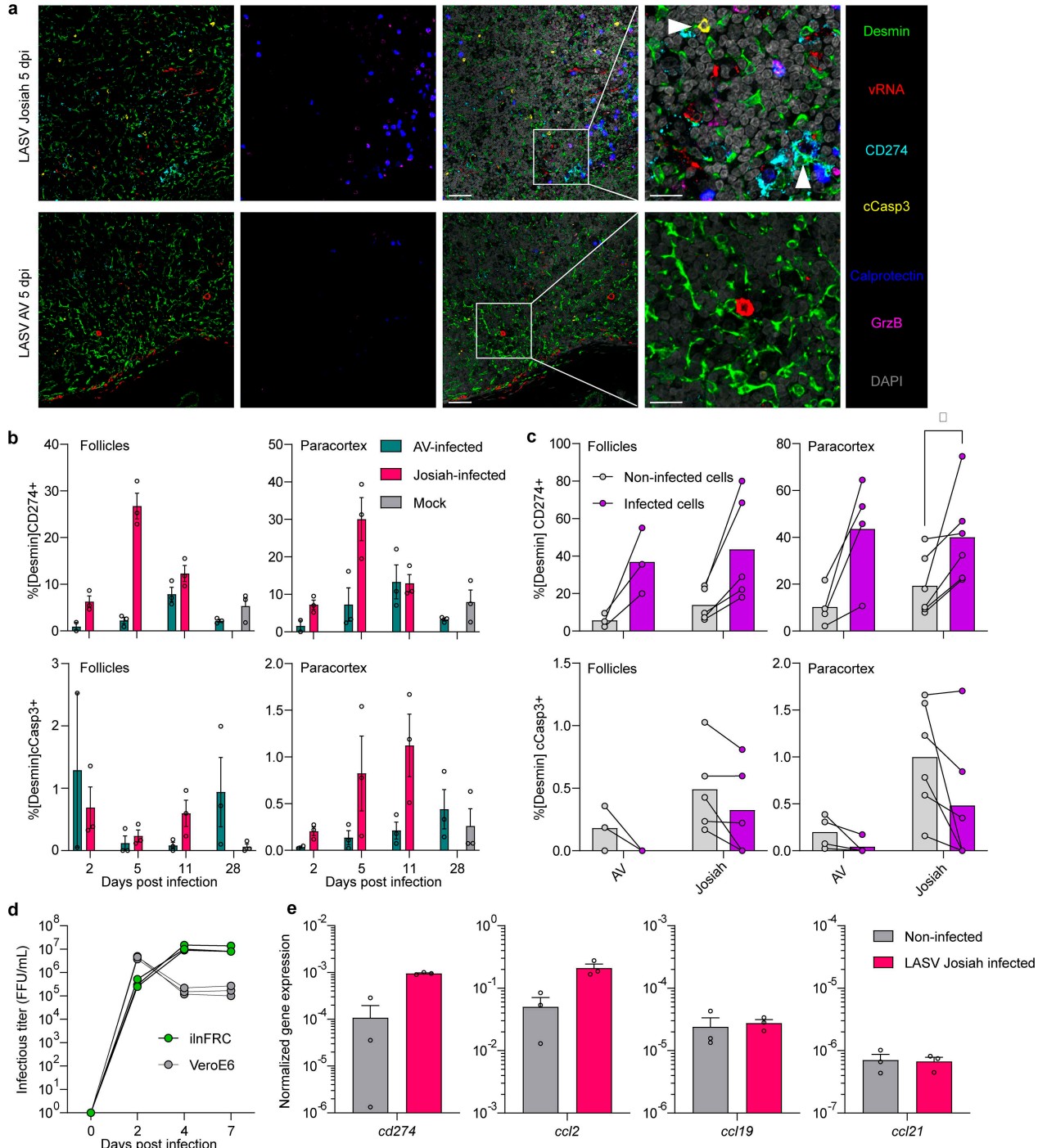

**Fig 5. Alterations of the stromal network of the LNs. A.** Confocal images captured from the paracortex of the LNs of infected animals showing stromal cells expressing CD274 (first column and vertical arrowhead in the fourth column), cCasp3+ stromal cells (first column and horizontal arrowhead in the fourth column), and neutrophils expressing GrzB (second column). First column: merge of the markers desmin, vRNA, CD274, and cCasp3; second column: merge of calprotectin and GrzB; third column: merge of all six markers with DAPI staining; fourth column: higher magnification of the white squares. Scale of columns 1-3 = 50 μm and column 4 = 20 μm. **B.** Percentage of CD274+ cells among total stromal cells (top row) and cCasp3+ cells among total stromal cells (bottom row) in the two main histological zones of the LN. Data and statistical considerations are as in Fig 5B-G. **C.** Comparison of the percentage positivity for CD274/ cCasp3 between infected and non-infected stromal cells for the AV and Josiah strains. A black line connects values originating from the same histological slide and the mean of each condition is represented. Organs harvested at 5 and 11 DPI were pooled

according to the viral strain and a minimum of 10 infected stromal cells was used as a cutoff to include an organ in the analysis, giving n = 3 AV-infected and n = 5 Josiah-infected. **D.** Kinetics of LASV Josiah replication in ilnFRCs or VeroE6 cells. Individual values corresponding to time points of the same experimental replicate (n = 3) are connected by a black line. **E.** Measurement of the expression of four relevant genes in infected ilnFRCs at 4 DPI by RT-qPCR. Values were normalized using the expression of the housekeeping gene GAPDH. Data are presented as individual values and the mean and SEM. No statistical test was appropriate to compare the infected and non-infected conditions.

this cell type was permissive to LASV infection and supported rapid and efficient viral replication, as infectious titers in the supernatant of these cells were higher than in VeroE6 cells after four days (Fig 5D). At 4 DPI, the mRNA encoding CD274 and CCL2 (MCP-1) was upregulated in infected cells versus non-infected cells, whereas the expression of the CCR7 ligand-encoding genes *ccl19* and *ccl21* was unchanged.

## Discussion

We took advantage of an NHP model of lethal versus non-lethal disease to explore the mechanisms responsible for the outcome of LASV infection. Data obtained on patients and NHPs, as well as during vaccination trials, have already suggested that cell-mediated immunity is key for protection against Lassa fever. Indeed, convalescent patients present a long-lasting, polyfunctional, and cross-reactive LASV-specific T-cell response [16–19,30], whereas the circulation of virus-specific antibodies does not correlate with survival or viral load in the blood [31] and neutralizing antibodies seem to appear only during the convalescence phase [22]. We demonstrate that survival seems to be associated in NHPs with the development of a LASV-specific T-cell response during the acute phase of the disease, whereas we did not detect such a response during lethal infection. Although a low percentage of LASV-specific T cells was detected 12 DPI in AV-infected animals, their presence in the blood suggests that effector T cells spread in infected tissues. An important role for LASV-specific T cells has also been suggested in vaccine studies in cynomolgus monkeys [32,33]. The adaptive immune response observed in AV-infected animals may allow the early control of LASV replication at the first infection sites and explains why the dissemination of this strain to peripheral organs was limited relative to the Josiah strain in our previous studies. The containment of viral replication probably prevents the uncontrolled and widespread innate immune activation induced by the massive release of viral products, as shown by the marked differences in cytokine levels between these two groups. We have previously studied the expression of several activation markers in the same PBMC samples and shown that a robust activation of CD4 + and CD8 + T cells was induced during the course of the disease in AV-infected animals but not during fatal infection [22]. As this study was not focused on LASV-specific T cells, it is probably that some of these cells were bystander T cells. This may explain the discrepancies observed with the current results in terms of percentage of responding cells and kinetic of response. Overall, these results suggest that the cell-mediated adaptive response plays a major role in the clearance of LASV and led us to explore the mechanisms responsible for the suppression of T cells in lethally infected animals. We show that the lethal Josiah strain, but not its non-lethal counterpart AV, preferentially infects immunoregulatory macrophages in the LNs at 5 DPI. These DC-SIGN-positive macrophages have been shown to be capable of producing high amounts of IL-10, inhibiting the activity of CD8+ T cells, and inducing the proliferation of Tregs [27,28]. No increase in the number of Tregs was nevertheless observed in the lymph nodes of infected animals. Immunoregulatory macrophages are also targets of infection for the Middle East respiratory syndrome coronavirus (MERS-CoV) and *Mycobacterium tuberculosis*, which allows these pathogens to replicate and disseminate while inducing a minimal immune response [29,34,35]. We infected monocyte-derived macrophages with LASV to determine whether the tropism for these cells is specific to LASV Josiah or whether it is infection by this strain that induces the differentiation of macrophages towards a regulatory phenotype. We previously showed that in-vitro infection of macrophages by LASV does not lead to the expected upregulation of co-stimulation and antigen presentation molecules and that infection of myeloid DCs leads to their activation but suppresses their ability to stimulate T cells in co-culture [36,37]. Here, we confirm that macrophage infection by both AV and Josiah strains activates canonical inflammation pathways but

induces downregulation of the MHC-II protein HLA-DR and overexpression of CD274. Although AV induced more intense and diverse modification of their transcriptome than Josiah, we failed to identify qualitative differences in the response between the two strains. These results do not exclude the possibility that, in vivo, macrophages respond qualitatively differently to Josiah and AV infection, as in-vitro, differentiated APCs do not always mimic the response induced by viral infections in these cells in vivo [36,37]. Indeed, cell-cell interactions and signaling by cytokines produced by other cell types cannot be replicated through in-vitro experiments. Moreover, one limitation of our model is the use of SC route to inoculate animals whereas natural LASV infection occurs through the respiratory tract. The first infected cells may differ from epithelial cells infected in the respiratory tract, and this could have consequences on innate immunity, as intracellular trafficking of arenaviruses and interactions with innate immunity sensors are tissue-specific and contribute to different host responses [38]. Hence, early tropism for immunoregulatory macrophages may allow LASV Josiah to delay local immune responses and spread to peripheral organs. Furthermore, we show that the progression of lethal infection is associated with a decrease in the frequency of mature monocytes in the blood, as well as upregulation of the expression of CD279 and a lack of HLA-DR expression by these cells. We also describe, for the first time, the expansion of MDSCs (predominantly of the granulocytic lineage) during fatal Lassa fever in the blood and their expression of CD279 and CD274. MDSCs make up a heterogeneous cell population that has been described during cancer and viral infections. They impede the T-cell response by secreting suppressive soluble mediators and expressing regulatory ligands [39–42]. They originate from immature myeloid progenitors released from the bone marrow and become activated peripherally. We and others have already shown that severe disease is associated with a cytokine storm, massive infiltration of immune cells in infected organs, and a transient drop in the frequency of circulating monocytes and NK cells [15,22,23,43]. Moreover, we have shown that Josiah infection leads to the circulation of a large proportion of immature neutrophils with a CD10⁻ phenotype [22,44]. Hence, we hypothesize that the overt inflammation observed in peripheral organs results in the depletion of immune cells in the blood, which could cause emergency myelopoiesis and the release of immature and suppressive myeloid cells. This phenomenon has been described during sepsis-induced immune dysfunction [45,46]. Indeed, we also observed the expansion of suppressive myeloid macrophages in the LNs, lungs, and intestine of infected animals and that these cells had close contacts with CD279-expressing T cells, suggesting that this suppressive pathway was effective and inhibited T cells during severe infection with LASV. The high levels of soluble PD-L1 and calprotectin detected at the peak of the disease in Josiah-infected animals confirmed the induction of immunosuppressive cells and the excessive activation of neutrophils. Indeed, the presence of sPD-L1 has been correlated with severity and immunosuppression during sepsis and SARS-CoV2 infection, and this soluble form can contribute to T-cell immunosuppression, particularly through apoptosis induction [47–50]. Calprotectin, the S100A8/S100A9 heterodimer, mainly derives from neutrophils and macrophages, is actively released during inflammatory process and contributes to excessive inflammation [51]. Calprotectin stimulates leukocyte migration by favoring leukocyte-endothelial cell interaction and increases vascular permeability. It has also proposed that in severe COVID19 patients, calprotectin accounts for the cytokine storm and may trigger the emergency myelopoiesis that leads to the release of immature and suppressive cells, which is consistent with our observations in Josiah-infected animals [52]. Bulk transcriptomic analysis of organs confirmed systemic activation of immunosuppressive pathways in the peripheral organs. Surprisingly, these pathways appeared to be less strongly upregulated in the LNs of Josiah-infected animals than those of AV-infected animals at 11 DPI, perhaps because the virus-specific T-cell responses present in the AV-infected animals induce physiological regulatory mechanisms. These observations suggest that suppressive myeloid cells play a key role in the inhibition of the T-cell response during lethal disease. This phenomenon may apply to other viral diseases, as studies on other VHFs caused by dengue virus and Bundibugyo ebolavirus have suggested that MDSC expansion is a common feature of these severe diseases [53,54]. Definitive proof of these pathogenic mechanisms would require ex-vivo confirmation of the capacity of these cells to inhibit T-cell activation and studies of the quantitative and qualitative functionality of hematopoiesis at late stages of the disease would provide important insights [55]. Altogether, these results provide an additional proof of evidence that the pathogenesis of severe Lassa fever is

related to viral sepsis. Indeed, most pathological events observed in our Josiah-infected animals are consistent with the definition of sepsis and septic shock that is a "life-threatening organ dysfunction caused by a dysregulated host response to infection" [56–58]. Multiorgan failure involving vascular leakage, liver and kidney failure, and acute respiratory distress characterized Lassa fever [8,22,23]. We have previously shown that a cytokine storm with dysregulated inflammatory and anti-inflammatory responses was induced in fatally-infected animals, and that their PBMC transcriptomic profiles were similar to the one observed during severe sepsis [22]. Although intravascular disseminated coagulation did not seem to occur in these animals, the coagulopathy observed was nevertheless close to the one described during sepsis [59,60]. The expansion of MDSCs and the immunosuppression, anergy, and apoptosis of T cells are also hallmarks of bacterial and viral sepsis [22,39,61–65]. Finally, we show that the stromal network of the LNs, in particular, FRCs, is an important target of replication for LASV and that their infection induces the upregulation of CD274 and CCL2, potentially contributing to the inhibition of T-cell activation and the chemotaxis of monocytes and neutrophils, respectively. The increased cell death in the stromal network, although apparently not a direct consequence of viral infection, may be involved in the loss of structure observed in the SLOs and, in turn, result in reduced interactions between APCs and lymphocytes and the loss of homeostatic cytokine gradients. Histological disorganization and infection of FRCs may also represent a common point between several VHFs, as this has already been suggested to occur during Ebola and Marburg virus infections [66]. According to these data, we propose a model for the pathological mechanisms underlying lethal disease, wherein LASV replicates in immunoregulatory macrophages in the LNs, which express low levels of antigen-presentation molecules, and induces a locally suppressive environment, impeding the immunological control of viral replication. Such immunological tolerance allows LASV to spread to the peripheral organs, where its replication leads to the massive release of viral products and induces an uncontrolled innate immune response and a cytokine storm. The translocation of circulating innate immune cells to the inflamed tissues leads to emergency myelopoiesis and the release of immature and suppressive myeloid cells that further inhibit the activation of LASV-specific T cells throughout the organism (S6 Fig). This hypothesis is consistent with previous data obtained elsewhere showing that fatal Ebola virus disease is characterized by elevated expression of CD279 and CD152 on circulating T cells [67]. It also suggests that residual immunosuppression may persist in convalescent patients who experienced a severe VHF, which may be linked to the observed excess mortality in convalescent Ebola virus disease patients [68]. Finally, these findings suggest that immune checkpoint inhibitors could be used to enhance the adaptive immune response in association with other therapeutics, such as antivirals, to limit viral replication, as well as molecules that target the inflammatory response to prevent deleterious innate immune activation, as suggested previously for viral sepsis [69]. However, even if the time between symptom onset and death is longer in humans than in our NHP model, the short disease course of Lassa fever could limit the benefit of such approach. Furthermore, as targeting some immune checkpoints such as PD1/PD-L1 may induce adverse effects or exacerbate inflammatory responses, it will be crucial to carefully evaluate these potential therapies in a relevant animal model. These therapeutic strategies could also be applied to other VHFs, as it appears that such pathological mechanisms are a common feature of this syndrome. In conclusion, we show that LASV-specific responses are key to controlling LASV and highlight how multiple immunosuppressive mechanisms lead to the anergy of T cells during lethal infection and overwhelming viral spreading.

## Materials and methods

### Ethics statement

Cynomolgus macaques were infected, monitored, sampled, and euthanized in a BSL-4 laboratory (Laboratoire P4 INSERM–Jean Mérieux) as described in Baillet et al. and all protocols were approved by the Rhône Alpes Ethical Committee for Animal Experimentation (file number 2015062410456662, CECAPP, UMS3444/US8, Lyon, France). All biological material used in this study has been previously obtained [22], except for the tissues of three additional mock-infected animals used for histological studies.

## Histology

Organs harvested at necropsy were fixed for two weeks in 4% formaldehyde, dehydrated, and embedded in paraffin. Three μm-thick slices were cut on a microtome before being processed on a Bond RxM staining station. Slides were dewaxed, $H_2O_2$ was added to quench endogenous peroxidases, and heat-induced antigen retrieval (HIER) was performed at 95°C for 30 min at pH 6 (ER1) or pH 9 (ER2). In certain cases, slides were incubated with a viral RNA-targeting probe (RNAscope, Biotechne) at 42°C for 2 h and then three amplification steps were carried out before pairing the adapters with horseradish peroxidase (HRP). In other cases, the incubation was first carried out with a primary antibody (see Table A in S1 Table) for 30 min at room temperature (RT) before adding a rabbit- and mouse-specific secondary antibody coupled to HRP. Fluorophores were then deposited on the target using a tyramide signal amplification system (Opal, Akoya Biosciences). Up to six cycles of immunodetection were repeated on the same slide using different fluorophores by stripping primary-secondary antibody complexes using HIER. Nuclei were then counterstained using narrow emission spectrum DAPI (spectral DAPI, Akoya Biosciences) and the slides were mounted in antifade mounting medium (Abcam). Images were captured on a confocal microscope (LSM980, Zeiss) using 9 nm-wide photodetectors to detect emitted light between 414 and 693 nm. For the LNs, the area of the slide was tiled with individual fields of view (FOV) to reconstitute a whole-slide image and for the lungs and intestine, 20 FOV were captured evenly across the organs (only in the epithelial layer for the latter). The signals associated with the various fluorophores were then deconvoluted using a linear unmixing algorithm based on the recognition of pre-recorded single spectra. Analysis was then performed with QuPath software using the following workflow: histological zones were manually annotated, individual cells were automatically detected using DAPI staining, a machine learning algorithm was trained to classify cell positivity for each marker, and cell-cell distances of each type were measured. To account for inter-slide variability in staining quality, classifiers were adjusted for each sample with blinding to the infection status of the animal. Results were then exported and processed using R Studio.

## Flow cytometry

Staining was performed on fresh peripheral blood for MDSC phenotyping or on cultured PBMCs/SLO cells for T-cell activation or on monocyte-derived macrophages. Cells were incubated with surface marker antibodies for 30 min at 4°C. For whole blood samples, red blood cells were lysed with the Immunoprep kit (Beckman Coulter). For other sample types, cells were permeabilized using a FoxP3 staining kit (Miltenyi) and non-specific binding sites were saturated using an FcR-blocking reagent (Miltenyi) before incubation with intracellular marker antibodies for 30 min at 4°C. Before analyzing the samples on the flow cytometer (Gallios 10-colors, Beckman-Coulter), cells were fixed in Stabilizing Fixative solution (BD Biosciences). Analysis was performed using Kaluza software (Beckman Coulter). The antibodies used can be found in Table B in S1 Table.

## T-cell activation assay

PBMCs were separated from whole blood samples using Ficoll density gradients and SLO cells were isolated using the protocol described in Baillet et al [22]. Samples were cryopreserved at -150°C before use. Samples were rapidly thawed, the DMSO-containing freezing medium was washed away, and the cells were left to rest for minimum of 4 h. The LASV-specific T-cell response was evaluated by stimulating cells with a pool of overlapping peptides spanning the Josiah or AV GPC and NP in the presence of anti-CD28 and anti-CD49d co-stimulatory antibodies (BD Biosciences), as previously described [70]. Staphylococcus enterotoxin A and phosphate-buffered saline (PBS) were used as positive and negative controls, respectively. Fourteen hours prior to staining for flow cytometry, cells were exposed to Brefeldin A (Sigma-Aldrich) at 10 μg/mL to block vesicular export. The percentage of cells responding to the antigenic stimulation was calculated by subtracting the percentage of positive cells in the non-stimulated condition from the percentage of positive cells in the stimulated condition.

### Infection of ilnFRCs

Human LN FRCs immortalized by transduction of papillomavirus genes E6/E7 (CancerTools) were exposed to LASV Josiah (GenBank HQ688674.1 and HQ688672.1) for 1 h at a multiplicity of infection (MOI) of 0.1 before washing away the inoculum. VeroE6 cells were used as a positive control of viral replication. The supernatant was harvested at 0, 2, 4, and 7 DPI, and cell lysates were harvested at 4 DPI for gene expression assays. Supernatants were titrated on VeroE6 cells as described in Baillet et al [22].

### Infection of monocyte-derived macrophages

Whole blood from healthy human donors was obtained through the Etablissement Français du Sang (EFS). PBMCs were isolated using Ficoll density gradients and subjected to Percoll density gradients to retrieve monocytes. Monocytes were further purified using a negative pan-monocyte selection kit with immunomagnetic separation (Miltenyi). Cells were then differentiated into macrophages in a medium containing 10% homologous plasma and 500 IU/mL recombinant human GM-CSF (Peprotech) for six days, with a renewal of 40% of the culture medium and 100% of the growth factor every 48 h. Plates were then put on ice for a minimum of 30 min and macrophages were mechanically detached using a cell scraper. Cells were exposed to LASV Josiah or LASV AV (GenBank FR832711.1 and FR832710.1) for 1 h at a MOI of 1 before washing away the inoculum. At 24 hpi, cell lysates were harvested for transcriptomic analysis, and at 48 hpi, cells were detached as previously described and stained for flow cytometry.

### RT-qPCR

Tissue homogenates were prepared by mechanical dissociation in a TissueLyser (Qiagen). Cell lysates from ilnFRCs were inactivated in RLT buffer (Qiagen). DNA was digested using a DNAse and total RNA was extracted using an RNeasy Mini kit (Qiagen), eluted in RNAse-free water, and stored at -80°C before processing. Nucleic acid was denatured and exposed to SuperScript III reverse transcriptase (FisherScientific) and the cDNA was mixed with primers and probes specific for genes of interest and DNA polymerase AmpliTaq Gold (FisherScientific) for amplification. The expression of the house-keeping gene GAPDH was used to normalize the results.

### Transcriptomic analysis

Tissue homogenates and cell lysates of macrophages were obtained and total RNA was extracted as previously described. The quality of RNA was confirmed using a TapeStation (Agilent) and RNA were quantified using a Nanodrop. mRNA was enriched from 100 ng of total RNA with NEXTFLEX Poly(A) Beads 2.0 kit (Revvity), then library preparation was performed with the MGIEasy RNA Directional Library Prep Set (MGI). Quality of libraries was confirmed using the Tapestation (Agilent) and quantified by Qubit 1X dsDNA HS Assay Kit (Invitrogen). Samples were pooled and after circularization in ssDNA, DNA nanoball sequencing was performed following the manufacturer's protocol on the MGI DNBSEQ-G400 (MGI), with 100 pb single-End and a large flow cell. The basecalling and the demultiplexing of data were done with BasecallLite v1.5.0.323. Bioinformatics analysis was performed using the RNAflow pipeline (https://gitlab.pasteur.fr/hub/rnaflow). Reads were cleaned of adapter sequences and low-quality sequences using cutadapt. Only sequences of at least 20 nucleotides in length were considered for further analysis. STAR, with default parameters, was used for alignment against the reference genome. Genomes were downloaded from UCSC and annotation track (.gtf Ensembl Genes) was retrieved from UCSC (genome: Crab-eating macaque, assembly Macaca_fascicularis_6.0 or human, assembly hg38). Genes were counted using featureCounts from the Subreads package (parameters: -t gene -g ID -O -s 2). Statistical analysis was performed using R software. For tissue samples, differential analysis was performed using the DESeq2 package. An independent DESeq2 model was independently run for each pairwise comparison. Groups were compared using a Wald test and p-values were adjusted using the Benjamini-Hochberg multiple testing correction.

For the macrophage study, genes with low read counts were filtered using the filterByExpr function of the edgeR package with the following parameters: min.count = 5, min.prop = 0.2, min.total.count = 10, and large.n = 50. Differential analysis was performed using the DESeq2 package. The model was adjusted for the infection status and the donor identifier. The local option was used to fit the dispersion trend. Groups were compared using the lfcShrink function with the "ashr" option and the p-values were adjusted using the Benjamini-Hochberg multiple testing correction. The GSEA analysis was performed using the CAMERA method of the limma package. The model was adjusted for infection status and patient identifier to account for the pairing, with the inter-gene correlation parameter set to 0. P-values were corrected using the Benjamini-Hochberg method. For single gene heatmaps, standardized gene expression was computed by normalizing the VST-transformed read counts against the control group and scaled to a standard deviation of 1. Genes were clustered using the Ward.D2 aggregation criterion and the Euclidean distance. The GSEA heatmaps show the enrichment Z-score computed by CAMERA for each comparison between two experimental groups. These heatmaps integrate the significance level of each gene set, as a grey color indicates nonsignificant gene sets with adjusted p-values > 5%. Gene sets were ranked according to their averaged Z-score for all comparisons.

### Detection of sPD-L1 and sCalprotectin

Soluble PD-L1 and calprotectin were quantified in plasma using human/cynomolgus monkey PD-L1/B7-H1 immuno-assay (DB7H10, R&D Systems Inc.) and monkey calprotectin (S100A8/S100A9) heterodimer ELISA kit (AEFI00469, AssayGenie), following manufacturer's instructions.

### Reagents

For information on the antibodies and probes used for the tissue and flow cytometry staining, as well as neutralizing anti-bodies, see S1 Table.

### Supporting information

**S1 Fig. Heatmap of the gene set "Immunosuppression".** Individual gene expression is averaged in each group, normalized by the non-infected condition, and color-coded according to the scale present on the right-hand side (n = 3 for each condition).
(TIF)

**S2 Fig. Heatmap of the gene sets "PD-L1 expression and PD-1 checkpoint pathway in cancer" and "Immunoreg-ulatory interactions between a lymphoid and non-lymphoid cell".** Individual gene expression is averaged in each group, normalized by the non-infected condition, and color-coded according to the scale present on the right-hand side (n = 3 for each condition).
(TIF)

**S3 Fig. Response of human macrophages to *ex-vivo* infection by AV or Josiah. A.** Comparison of the percentage of CD274+, CD14+ HLA-DR-, and CD163+ macrophages in infected and non-infected conditions. A black line connects values obtained from the same human donor and the mean of each condition is presented. According to the result of a D'Agostino-Pearson normality test, we performed a paired one-way ANOVA with Tukey's test or a non-parametric Fried-man's test with Dunn's test for multiple comparisons. **B.** Volcano plot showing the transcriptomic modifications induced by AV or Josiah infection in macrophages. Each gene is represented by a point, blue for an adjusted p-value < 0.05, red for an adjusted p-value < 0.05 and a |log fold change| > 1, or gray when none of the conditions are met. **C.** Boxplots of the expression of several gene sets relative to the non-infected value. Data presentation and statistical analysis are as in S1 Fig, adding the representation of individual genes in gray circles. **D-E.** Heatmaps of the expression of gene sets

"Cytokines" and "Immunosuppression". Data are averaged according to the viral strain, normalized by the non-infected value, and color-coded. For the "Cytokines" gene set, all genes shown are significantly differentially expressed (sDE) between at least one of the viral strains and the non-infected condition are presented, whereas all genes of the "Immuno-suppression" gene set are presented and sDE genes are labeled by an asterisk. **F.** Heatmaps of the gene set enrichment score of canonical pathways from the HALLMARK and REACTOME gene sets collections. Only pathways that are significantly different for at least two comparisons are shown. For **A** to **F**, data represent four human donors.
(TIF)

**S4 Fig. Histological study of Tregs in the LNs.** Confocal images illustrating the quantification presented in Fig 5F, allowing the visualization of Tregs (CD3+ FoxP3+, first column) and an unidentified cell positive for IL-10 mRNA (second column and white arrowhead in the fourth column). First column: merge of the markers CD3 and FoxP3, second column: merge of Ki67 and IL-10 mRNA, third column: merge of all four markers with DAPI staining, fourth column: higher magnification of the white squares. Scale of columns 1–3 = 50 μm and column 4 = 20 μm.
(TIF)

**S5 Fig. Histological study of suppressive macrophages in peripheral organs.** Confocal images illustrating the quantification presented in Fig 5G, allowing the visualization of CD68+ CD274+ HLA-DR- cells in the epithelial layer of the large intestine and the lung. First column: merge of the markers CD68 and HLA-DR, second column: merge of CD68 and CD274, third column: merge of all three markers with DAPI staining, fourth column: higher magnification of the white squares. Scale of column 1-3s = 50 μm and column 4 = 20 μm.
(TIF)

**S6 Fig. Model of the immunosuppressive mechanisms leading to T-cell anergy in fatal Lassa fever.** During fatal infection, the virus replicates rapidly in immunoregulatory macrophages, inducing defective priming of cellular adaptive immunity and allowing LASV to spread to peripheral organs and infect them. This pantropism induces the massive release of pathogen-associated molecular patterns (PAMPs), the recruitment of innate immune cells, and uncontrolled immune activation. Emergency myelopoiesis is initiated to replace the circulating pool of innate immune cells, leading to the release of immature myeloid cells that accumulate in the organs and further suppress the specific T-cell response. In parallel, alterations of the SLO stromal network impede T-cell homeostasis and activation, allowing relentless replication of the virus. Continuous contact with immunosuppressive myeloid cells and regulatory cytokines culminates in the exhaustion of primed T-cell clones and T-cell anergy. Created using BioRender under the CC BY 4.0 Licence.
(TIF)

**S1 Table.** **(A)** Antibodies and fluorophore panels used for histological studies. HIER buffer ER1 has a pH of 6 and ER2 has a pH of 9. Incubations with all primary antibodies were carried out for 30 min at RT, except CD279 for which the incubation was carried out for 1 h at RT. **(B)** Antibodies used for flow cytometry panels. Intracellular markers are in shown blue.
(DOCX)

**S1 Data. xlsx file contains all numerical data of each figure of the study.**
(XLSX)

## Acknowledgments

We thank J. Brocard, E. Chatre, and E. Caracas Bobocioiu (Plateau Technique Imagerie/Microscopie, SFR Biosciences, UAR3444 Centre National de la Recherche Scientifique, US8 Institut National de la Santé et de la Recherche Médical, Ecole Normale Supérieure de Lyon, Université Claude Bernard Lyon 1) for their help with the multispectral confocal microscopy. We are grateful to S. Becker for providing us with the LASV strain Josiah. We thank B. Renaudin for her

administrative support and the Jean Mérieux – INSERM BSL4 team for their help and for the animal experiments. We thank F Relouzat and R Le Grand (CEA, Fontenay-aux-Roses) for technical help. We thank N. Baillet for setting up the initial animal experiments. We thank the Etablissement Français du Sang and blood donors for providing blood from healthy donors.

## Author contributions

**Conceptualization:** Joël Lachuer, Emeline Perthame, Sylvain Baize.

**Formal analysis:** Blaise Lafoux, Natalia Pietrosemoli, Emeline Perthame, Sylvain Baize.

**Funding acquisition:** Sylvain Baize.

**Investigation:** Blaise Lafoux, Gustave Fourcaud, Jimmy Hortion, Laura Soyer, Alexandra Journeaux, Clara Germain, Stéphanie Reynard, Hadrien Cousseau, Clémentine Larignon, Natalia Pietrosemoli, Séverine Croze, Joël Lachuer, Emeline Perthame, Sylvain Baize.

**Methodology:** Sylvain Baize.

**Project administration:** Sylvain Baize.

**Supervision:** Sylvain Baize.

**Validation:** Sylvain Baize.

**Writing – original draft:** Blaise Lafoux.

**Writing – review & editing:** Sylvain Baize.

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
