## [Decision Letter · Decision Letter 0]

11 Mar 2025

PPATHOGENS-D-25-00132

Expansion of myeloid suppressor cells and suppression of Lassa virus-specific T cells during fatal Lassa fever

PLOS Pathogens

Dear Dr. Baize,

Thank you for submitting your manuscript to PLOS Pathogens. After careful consideration, we feel that it has merit but does not fully meet PLOS Pathogens's publication criteria as it currently stands. Therefore, we invite you to submit a revised version of the manuscript that addresses the points raised during the review process. Both reviewers had concerns that some data from the same animal experiment had been previously published. Please ensure that it is crystal clear what was previously analyzed and which data are entirely new. Reviewer 1 had specific questions about discrepancy between the time points noted in the previous study (Baillet et al, 2021) and the time points used in this study. This reviewer also had other concerns regarding whether the rigor of the data presented support the conclusions made (Point #1-7). Please answer all the reviewers' pertinent queries although I believe some (e.g. point #4, reviewer 1) is a matter of language clarity. Reviewer #2 had concerns about over-interpretation of the data that the authors should address.          

Please submit your revised manuscript within 60 days May 10 2025 11:59PM. If you will need more time than this to complete your revisions, please reply to this message or contact the journal office at plospathogens@plos.org. Please include the following items when submitting your revised manuscript:

We look forward to receiving your revised manuscript.

Kind regards,

Benhur Lee

Section Editor

PLOS Pathogens

 Sumita Bhaduri-McIntosh

Editor-in-Chief

PLOS Pathogens

orcid.org/0000-0003-2946-9497 Michael Malim

Editor-in-Chief

PLOS Pathogens

orcid.org/0000-0002-7699-2064

**Journal Requirements:**

2) We noticed that you used the phrase 'data not shown' in the manuscript. We do not allow these references, as the PLOS data access policy requires that all data be either published with the manuscript or made available in a publicly accessible database. Please amend the supplementary material to include the referenced data or remove the references.

3) Thank you for including an Ethics Statement for your study. Please include:

i) The full name(s) of the Institutional Review Board(s) or Ethics Committee(s)

ii) The approval number(s), or a statement that approval was granted by the named board(s)

iii) A statement that formal consent was obtained (must state whether verbal/written) OR the reason consent was not obtained (e.g. anonymity). NOTE: If child participants, the statement must declare that formal consent was obtained from the parent/guardian.].

5) We notice that your supplementary Figures, and Table are included in the manuscript file. Please remove them and upload them with the file type 'Supporting Information'. Please ensure that each Supporting Information file has a legend listed in the manuscript after the references list.

Potential Copyright Issues:

i) Figure S5: We noted that the figure is created using BioRender. Please confirm that you hold a Premium account and provide a pdf copy of the CC BY 4.0 Licence as provided by BioRender. For instructions on how to generate a CC BY 4.0 license for your figure, please see the guidelines here: https://help.biorender.com/hc/en-gb/articles/21282341238045-Publishing-in-open-access-resources. 

If you are using the free assets from BioRender, we are unable to publish these images as they are licenced under a stricter licence than CC BY 4.0. In this case we ask you to remove the BioRender images and replace them with open source alternatives.

See these open source resources you may use to replace images / clip-art:

- https://openclipart.org/

7) Please amend your detailed Financial Disclosure statement. This is published with the article. It must therefore be completed in full sentences and contain the exact wording you wish to be published.

**Comments to the Authors:**

**Please note that one of the reviews is uploaded as an attachment.**

**Reviewers' Comments:**

Reviewer's Responses to Questions

**Part I - Summary**

Reviewer #1: The aim of the current study is to provide evidence that recovery from acute LF is associated with LASV-specific CD4+ and CD8+ T-cell responses. The authors used frozen samples collected during 2021 Baillet N et al study to stimulate PBMC with a pool of overlapping peptides spanning the Josiah or AV GPC and NP in the presence of co-stimulatory antibodies and to count cytokine-secreting cells by flow cytometry. They found a gradual increase in the frequency of LASV-specific CD4+ and CD8+ T cells at 12 dpi and culminating at 32 dpi (Fig. 1A). By contrast, they did not observe any increase in the virus specific T-cells in Josiah-infected macaques. Using multiplexed immunofluorescence (IF) and in-situ hybridization (ISH) of histology sections of inguinal LN, the authors presented evidence that LASV Josiah infection was predominantly associated with DCs and immunoregulatory macrophages, while the AV strain was detected mostly in macrophages, endothelial, and stromal cells (Fig. 2). Transcriptomic analysis of the organs of infected animals and human PBMC-derived macrophages infected in vitro with Josiah and AV strains revealed more gene expression alterations during AV infection vs Josiah infection. Based on presented data, the authors hypothesized that bold inflammation observed in peripheral organs of Josiah-infected animals results in the depletion of mature monocytes and the expansion of immunosuppressive myeloid cells, alterations of LNs, and suppression of the virus-specific T-cell responses. In sum, presented data provides incremental contribution to better understanding of LF pathogenesis.

Reviewer #2: This article describes the immune response in a simian model of Lassa virus infection. This work complements two previously published articles by the same team on the same animals: Baillet et al., Comm Biol 2021, and Hortion J, Pos Pathog 2024. Here, they specifically describe the immunosuppressive aspect of this response, particularly in animals that do not survive in the lethal model (Josiah). The experiments and techniques used are appropriate, and the results obtained are thus clear-cut, allowing for a straightforward analysis of the findings. These results are clearly of great interest as they indicate the emergence of elements suggestive of immunosuppression in the group of non-surviving animals: a decrease in specific T cells, systemic dissemination of this immunosuppression (no compartmentalization), the appearance of MDSCs, immature neutrophils and HLA-DRlow monocytes, overexpression of PD-1 and PD-L1, and the disappearance of non-classical monocytes. As such, they deserve to be highlighted. That said, I have several remarks.

**Part II – Major Issues: Key Experiments Required for Acceptance**

Reviewer #1: Major concerns:

1. Study design. Sample collection procedures are poorly described. It looks like the most valuable samples for measuring LASV-specific T cell responses and histology (IF, ISH) analysis were collected during 2021 study (Baillet et al.). In this study, the latest point was 28 dpi. In the current study, the samples collected at 32 dpi were also analyzed. Are these late samples from the same or from “different” study? The authors must present clearly description of all samples used in this study with timing information and quality assessment.

The authors used up to six cycles of immunodetection with same tissue sections by repeatedly stripping primary-secondary antibody complexes. Preservation of original antigen epitopes in heavily used tissue sections must be properly validated to exclude false-staining results. Why 6 cycles? Are staining patterns after two stripping procedures the same as after 6 cycles?

2. While the cell-mediated adaptive response plays a major role in the clearance of LASV during natural infection and/or vaccination, the data presented by the authors does not support this concept. The T cell kinetics data in Baillet et al (2021) publication and in current study (Fig. 1) are different and contradictory. In the Baillet et al (2021) study (Fig. 6), CD4+ and CD8+ T cells expressing functional markers of immune activation were picked at 12-15 dpi in AV-infected monkeys and declined at 28 dpi. These T cell response kinetics are in line with assumption that T cell responses play a major role in controlling viremia (picked on day 12). In the current study (using the same PBMC samples?), LASV AV-specific T cells were only barely detected on day 12 “and culminating at 32 dpi” (see below). As in case of neutralizing antibodies, these LASV-specific T cell responses seem to appear too late and in limited numbers to contain LASV replication. These contradictory T-cell results must be properly addressed and discussed.

Reviewer #2: (No Response)

**Part III – Minor Issues: Editorial and Data Presentation Modifications**

Reviewer #1: Minor concerns:

3. The NHP is the best experimental model to study LF pathogenesis. The route of the infection plays an important role in pathogenesis. Subcutaneous injections poorly mimic the natural LASV transmission. The authors must clearly articulate deficiency of their animal model. It is difficult to consider the inguinal LN as “the first site of infection” (see page 7, the last sentence). In fact, epithelial cells of the respiratory and/or digestive tracts are among the first host cells to interact with the virus during natural transmission/infection. Interaction with polarized epithelial cells is host-tissue and virus-strain specific and differently contributed to induction of innate immunity and LF pathogenesis. Experiments in NHPs provide evidence that interaction with mucosa of gastrointestinal tract and/or crossing epithelium barrier contributes to virus attenuation. Likewise, intracellular trafficking and interactions with innate immunity receptors are also strain-specific events that contribute to different patterns of innate immune responses and pathogenicity (e.g. Jonhson D et al., 2024). All these issues must be properly discussed.

4. Page 7. At the middle of this page the authors noted, “a strong and gradual increase in the frequency of LASV-specific T cells …, starting at 12 dpi and culminating at 32 dpi (Fig. 1A)”. This notice is not correct. Since there are no samples/data available after 32 dpi, there is no experimental basis to support “culminating” claim. It is also incorrect regarding “gradual increase”, since LASV AV-specific CD8 T cells declined at 18 dpi in comparison with 12 dpi.

5. The same page, last sentence. The authors stated that LN is the first site of infection where the immune response is primed. It suggests that LASV AV infection must be associated with favorable immune regulatory profile in the peripheral LN. However, as seen in Fig. 2d, immunosuppressive genes were strongly upregulated in AV infected LN vs samples collected from solid tissues.

6. It is also not clear why measurement of immunosuppressive “relevant genes in different organs” was limited to LN, intestine, and lung. Why did not measure in liver and spleen to keep at least partial consistency with data presented in panel D, Fig. 2?

7. Page 12. The authors stated that “survival is associated in NHPs with the development of a LASV-specific T-cell response during the acute phase of the disease, whereas we did not detect such a response during lethal infection”. However, AV-specific T cell responses were only barely detected at 12 dpi and CD8+ T cells and declined at 18 dpi. There is no evidence that limited AV-specific CD8+ T cells really contribute to the viral control.

Reviewer #2: The most remarkable aspect of this study is the similarity of the results to those described in septic shock in human clinical cases. However, this aspect is only briefly mentioned, despite offering a major perspective for understanding the pathophysiology of severe infections, regardless of their origin. Here, the concept of viral sepsis should be considered. Numerous clinical references on bacterial sepsis should be added to the discussion to support this point, as this would reinforce the demonstration.

In general, the manuscript should focus on the key findings that illustrate this similarity with septic shock. Too many results are presented, which dilute the main message.

In this context, the results in Figure 3 do not contribute to the demonstration and should be removed.

Do the authors have, among the numerous collected data, any results that could highlight an increase in regulatory T cells?

The authors should be more cautious in their conclusions, as they do not establish a causal link between immunosuppression and the death of monkeys. The emergence of immunosuppression may simply be an indicator of an excessive inflammatory response (see septic shock mechanisms). In septic shock, immunosuppression is responsible for late mortality (i.e., after the first few days) by promoting nosocomial infections and prolonging ICU stays. However, here, no animal survives beyond day 12, making it difficult to extrapolate long-term impacts. While immunosuppression is likely partially detrimental, it is probable that mortality is primarily due to organ failure related to the inflammatory response, which in turn reflects the viral load progression in the early days (as described in Baillet et al.). The absence of medium-term survivors in the Josiah group is a limitation of the study that should be acknowledged.

In connection with this last point, the timeline of mortality in Lassa fever clinical cases should be recalled to consider potential therapeutic options for survivors.

The authors' proposal to use anti-PD-1 may seem contradictory to certain recommendations advocating the use of anti-inflammatory treatments. How can these different approaches be reconciled?

PLOS authors have the option to publish the peer review history of their article (what does this mean? ). If published, this will include your full peer review and any attached files.

**Do you want your identity to be public for this peer review?** For information about this choice, including consent withdrawal, please see our Privacy Policy .

Reviewer #1: **Yes: ** Igor S. Lukashevich

Reviewer #2: No

**Figure resubmission:**
---

## [Editor Report · Decision Letter 1]

8 Apr 2025

Dear Dr. Baize,

We are pleased to inform you that your manuscript 'Expansion of myeloid suppressor cells and suppression of Lassa virus-specific T cells during fatal Lassa fever' has been provisionally accepted for publication in PLOS Pathogens.

Best regards,

Benhur Lee

Section Editor

PLOS Pathogens

Benhur Lee

Section Editor

PLOS Pathogens

Sumita Bhaduri-McIntosh

Editor-in-Chief

PLOS Pathogens

orcid.org/0000-0003-2946-9497

Michael Malim

Editor-in-Chief

PLOS Pathogens

orcid.org/0000-0002-7699-2064

Reviewer Comments (if any, and for reference):

The authors have made a good faith effort in answering the reviewers' comments and have more transparently discussed the limitations of the model. Conclusions are qualified to what the data shows.

---

## [Editor Report · Acceptance letter]

Dear Dr. Baize,

We are delighted to inform you that your manuscript, "Expansion of myeloid suppressor cells and suppression of Lassa virus-specific T cells during fatal Lassa fever," has been formally accepted for publication in PLOS Pathogens.

Best regards,

Sumita Bhaduri-McIntosh

Editor-in-Chief

PLOS Pathogens

orcid.org/0000-0003-2946-9497

Michael Malim

Editor-in-Chief

PLOS Pathogens

orcid.org/0000-0002-7699-2064